# Improving residents' satisfaction with administrative boundary changes: A comparative analysis based on the township-town merger policy

Qiong Wang[1], Hao Zhao[2]*, Jie Sun[1], Yajie Yu[1], Changlin Zhang[1], Pingfan Hao[1]

1 College of Foreign Languages, Nanjing Xiaozhuang University, Nanjing, Jiangsu, China, 2 College of Marxism, Hainan University, Haikou, Hainan, China

* dudu87@hainanu.edu.cn

## Abstract

Administrative boundary changes driven by policies are pivotal instruments for improving resource allocation efficiency. Studying the factors driving residents' satisfaction with these policies is crucial for enhancing policy effectiveness. Based on field surveys in Lukou and Hengxi Subdistricts, this paper constructs a logit model to identify key determinants of residents' satisfaction with the township-town merger policy (TMP) and applies the Fairlie decomposition to quantify the sources of satisfaction disparities between subdistricts. Results indicate that employment incentives, social security, and cultural development are the primary drivers of policy satisfaction and interregional differences, exerting greater influence than individual demographic traits. Mechanistically, the study reveals that the development approaches of different regions are strongly associated with policy satisfaction. Besides, residents in communities with stronger collective identity tend to evaluate policy outcomes through temporal comparisons with their own past, leading to higher satisfaction when improvements in social security and cultural preservation are evident. Conversely, in areas with weakened collective identity, residents form higher expectations through upward social comparisons with more developed neighboring subdistricts. When actual policy outcomes fall short of these expectations, satisfaction declines. These findings offer a micro-perspective on how top-down administrative boundary reforms affect public satisfaction, providing recommendations for policy design, sustainable governance, and regional planning.

## Introduction

Administrative boundary changes (ABCs), as a fundamental tool of spatial and territorial governance, represent a significant form of state intervention in reshaping economic geography and social landscapes [1–4]. These changes can emerge

**Data availability statement:** All relevant data are within the paper and its Supporting information files.

**Funding:** The author(s) received no specific funding for this work.

**Competing interests:** The authors have declared that no competing interests exist.

**Abbreviations:** TMP, Township-town merger policy; ABC, Administrative boundary change.

either as part of top-down territorial reforms or be driven from the bottom up by local authorities or residents [2]. The former relies on government regulation, often involving mandatory consolidation initiated and enforced by higher-level governments in accordance with laws and regulations [3]. This typically takes the form of policies issued by the central government to merge [5] or split [6] multiple administrative units. Bottom-up boundary changes, on the other hand, are often enacted through public referendums, emphasizing civic will and serving as a major driver of such adjustments [2].

A wave of boundary changes, which is manifested mainly as municipal mergers, emerged in the latter half of the 20th century, spanning numerous countries worldwide [7]. Such reforms are typically characterized by a degree of local deliberation and public participation, albeit not without contention [2,7,8]. In contrast, boundary adjustments within China's political system, most notably the township-town merger policy (TMP), are predominantly top-down, strategic initiatives driven by the imperatives of urbanization and macroeconomic restructuring. China's TMP offers a typical case for studying administrative boundary changes. As a state-led initiative spanning over 70 years and affecting tens of thousands of towns and villages, it has exerted profound impacts. However, its efficiency-oriented logic, while effectively reorganizing local governance and economic space, has often overlooked grassroots community identity and the accessibility of public services, thereby triggering resident dissatisfaction. Therefore, studying this policy provides critical insights into the inherent tensions between top-down political-economic objectives and bottom-up social well-being during boundary changes, and offers a basis for comparative analysis with other, more participatory municipal merger models worldwide.

More precisely, the TMP falls under the common ABC types of "merger" and "splitting." Its policy evolution has undergone four key phases: initial standardization and adjustment (1949–1961), stagnation and contraction (1962–1983), rapid expansion (1984–2000), and quality enhancement (2001-present). This process clearly reflects how administrative divisions have shaped boundary changes at the grassroots level in China, illustrating a strategic shift toward structural optimization and enhanced governance capacity. Scholarly research on township governance, particularly since the reform era, has documented these institutional transformations [9].

Overall, the TMP has indeed generated substantial socio-economic benefits for local populations, fostering administrative efficiency, economic agglomeration, and the advancement of urbanization [10–12]. Nevertheless, existing studies have raised doubts regarding its overall effectiveness. The growing burden of education costs, the erosion of local civic leadership, the expansion of service radii, the marginalization of certain economies, and the disruption of cultural continuity have all contributed to skepticism toward these administrative boundary adjustments [13,14].

In this context, examining residents' satisfaction with TMP becomes particularly vital. Since the township-town merger policy adopts a top-down implementation approach, investigating residents' satisfaction can help compensate for its lack of resident participation. As the primary beneficiaries of the policy, residents' satisfaction

serves as a key indicator of its effectiveness, representing a fundamental prerequisite for the sustainable development of merged towns and optimizing the overall outcomes of administrative boundary reforms.

Consequently, the research questions addressed in this paper are as follows: (1) What factors influence residents' satisfaction with the policy? (2) What factors account for the disparity in policy satisfaction between residents of the two Subdistricts? (3) What underlying mechanisms explain these differences? Specifically, this study selects the geographically adjacent Lukou and Hengxi Subdistricts in Nanjing City for comparative analysis to explore the driving factors behind residents' satisfaction with the TMP. A Logit regression model is employed to examine the determinants of policy satisfaction, encompassing both individual characteristics and policy-related effects. Furthermore, the Fairlie decomposition method is applied to disentangle the differences in policy satisfaction between the two Subdistricts and to quantify the contribution and significance of each variable. Combined with qualitative evidence drawn from resident interviews, the study further investigates the underlying causes of these disparities.

By addressing these issues, this paper contributes to the existing literature in three important ways: (1) While previous studies have primarily examined the macro-level impacts of the TMP on economic and social development, this study focuses on policy effects through a micro-level lens of residents' satisfaction, which reflects the changes brought by boundary adjustments. Moreover, existing research has largely emphasized the direct economic impact of the strategies, overlooking the influence of non-economic factors. This study fills a gap by exploring the essential dimensions of inclusive growth, including social security and cultural heritage.

(2) When examining the policy effects of the TMP, previous literature has often regarded it as a uniform policy exerting equal influence across regions, giving little attention to potential variations in its impacts or the underlying sources of such differences. Surprisingly, this study reveals that residents' policy satisfaction continues to diverge markedly even two decades after the implementation of the TMP, despite the two regions being geographically, economically, and culturally similar. Based on quantitative Fairlie decomposition and qualitative interview analyses, this paper delves deeper into the reasons for these disparities, discovering that comparative patterns vary across regions and that identity recognition plays a significant role in enhancing policy satisfaction. By doing so, the study provides valuable insights for the context-specific and adaptive design of TMP and other upper-level policy initiatives.

(3) The findings of this study collectively contribute a Chinese case to the global research on administrative boundary changes. By conceptualizing the township-town merger as a form of mandatory administrative boundary adjustment, this research elucidates the multifaceted influence of top-down boundary restructuring on residents' economic and social lives through the lens of their policy satisfaction. It offers a more contextualized, resident-centered analytical perspective that addresses the limitations of previous studies, which often relied on overly broad approaches. The conclusions drawn here enhance the understanding of state-led boundary reorganization and may help increase public support for sustainable land governance in countries with a similar administrative structure to China.

## Literature review

### Determinants of policy satisfaction

According to Diener [15] and Pelletier [16], "Satisfaction represents an evaluation of the congruence between a person's life experience and some particular standard." Existing research on policy satisfaction predominantly draws on customer satisfaction theory, which posits that satisfaction is determined by the comparison between customers' expectations of a product or service and their actual experiences [17]. Applying this framework, scholars conceptualize citizens as "customers" of public services, with policy satisfaction representing the public's perception of the quality of a policy as a "public product." The central premise is that public satisfaction is jointly shaped by expectation levels and perceived policy outcomes. In other words, satisfaction increases when the actual effects of a policy exceed expectations, and conversely, it declines when outcomes fall short [18].

Public satisfaction is a crucial element and an instrument for evaluating government performance, serving as a key indicator of policy effectiveness [19]. Existing research has examined the determinants of policy satisfaction at both the individual and policy levels.

At the individual level, personal and household characteristics, as well as subjective perceptions, collectively shape policy satisfaction. Specifically, demographic factors such as education, age, employment status, and income significantly influence evaluation tendencies. For instance, Clifton et al. [20], in their study of utility policies across 12 European countries, found that individuals with lower educational attainment expressed markedly lower satisfaction with telecommunications and energy policies than those with higher education. Non-employed individuals, being more sensitive to the economic benefits of policies, displayed satisfaction levels closely tied to cost-related policies, such as telecommunication tariffs, which were readily affected by expenditure pressures. Elderly groups demonstrated higher satisfaction with security-oriented policies, including poverty alleviation and health welfare [21], yet their weaker adaptability to technology-driven policies resulted in relatively lower satisfaction in this domain. In contrast, younger populations exhibited greater satisfaction with development-oriented policies, such as ecological protection and entrepreneurship support [22]. Households with higher income levels also tended to show greater openness to and approval of policies [23].

In terms of subjective perception, residents' psychological policy identification, degree of information mastery, and perceived policy quality constitute core determinants of satisfaction. Zhang et al. demonstrated that both policy identification and the extent to which residents master policy-related information are necessary conditions for achieving high levels of policy satisfaction [24]. Gu found that farmers' perceptions of policy fairness exerted a substantially greater influence on satisfaction than any other factor [25]. Moreover, residents' recognition of their identity profoundly influences policy satisfaction. When people positively acknowledge the identity transformation triggered by policies, satisfaction levels rise markedly. Conversely, they decrease. Research further indicates that identity recognition doesn't operate in isolation. Instead, it interacts with factors such as policy perceptions, income levels, and social relationships to directly or indirectly affect satisfaction [26,27].

At the policy level, existing research underscores that policy implementation serves not only as a crucial step toward achieving policy objectives but also as a measure of the scientific rigor and validity of policies [28]. The direct benefits individuals experience and the effectiveness perceived during policy implementation constitute direct determinants of satisfaction, serving as an intuitive reflection of policy outcomes.

More specifically, first, the direct economic benefits conferred by policies represent a core driver of satisfaction [29]. In the context of social security policies, both the breadth of coverage and the level of protection directly shape policy satisfaction [30] and further influence satisfaction with derivative policies such as rehabilitation benefits [21]. Second, the effectiveness of policy implementation, which encompasses the transparency of policy information and the fairness of procedural execution, exerts a substantial impact on satisfaction [23,31]. Third, the adaptability of policies, also known as the extent to which policy design aligns with individual needs, emerges as another direct determinant of satisfaction. For instance, the ecotourism policy tailored to the livelihood transition needs of residents in Wuyishan National Park achieved high satisfaction, demonstrating that enhancing policy satisfaction requires precise tailoring by governments [22]. Additionally, policy satisfaction reflects a trade-off between short-term and long-term benefits, in which residents' confidence in the policy's long-term advantages, coupled with the belief that these benefits compensate for short-term deficiencies, significantly enhances overall satisfaction [26].

## The multifaceted impact of township-town merger policy

Many countries worldwide, particularly those experiencing rapid urbanization, have implemented the TMP with the objective of optimizing grassroots administrative structures and enhancing governance efficiency. The TMP facilitates the consolidation of administrative frameworks and improves governance effectiveness through coordinated planning and unified service provision [12], thereby reducing the costs associated with the allocation of resources [11]. Nevertheless, structural

challenges arising from policy implementation have also intensified local governance difficulties. During the transitional period, ambiguities in the division of responsibilities and limited fiscal autonomy have undermined the capacity for coordinated governance. Merged towns frequently face issues such as unclear administrative boundaries, reduced service functions, and underutilized resources, leading to misalignment with residents' needs [32]. For instance, the mergers have increased the distance between government authorities and local populations, consequently limiting citizens' opportunities for participation in public decision-making [33].

In addition to its influence on governmental governance, existing literature emphasizes that policy implementation represents a crucial stage for realizing policy objectives and functions as a benchmark for evaluating the scientific rigor of policies [28]. Scholars have assessed the outcomes of the TMP through the dimensions of economic and social development [32,34]. These outcomes directly shape residents' perceived benefits and constitute a pivotal factor in determining policy satisfaction, although scholarly perspectives on their effects remain contested. At the level of economic growth, some studies affirm that township-town merger policies can exert positive effects on local economies by fostering agglomeration economies, enhancing productivity, and achieving spatial and demographic economies of scale [12]. Within the context of China's county-to-district mergers, such consolidations have generated short-term boosts to urban economic growth, primarily driven by large-scale infrastructure investments [34]. Moreover, research indicates that boundary adjustments can narrow economic gaps [3] and even affect environmental outcomes [35].

However, a contrasting body of evidence underscores notable negative economic consequences, particularly for merged units situated on the political periphery. When county-level entities are transformed into city districts, they often lose control over fiscal resources, land-use planning, and local development strategies, leading to declines in resident savings, average wages, and long-term macroeconomic prospects in the affected areas [34]. The model in which strong townships absorb weaker ones, without accompanying substantive economic transformation, further illustrates the potential risks inherent in such mergers [32].

At the level of social development, prior research examining administrative mergers has highlighted significant shifts in socio-spatial dynamics. A frequently observed outcome is the agglomeration of population and labor resources, driven by the expansion of built-up areas and industrial upgrading. The enlargement of urban space enables governments to intensify investments in residential construction and public infrastructure [34], thereby improving the accessibility and quality of public services, medical care, and healthcare resources throughout the process of administrative division adjustments [36]. Nonetheless, these benefits are often unevenly distributed. There exists a risk that infrastructure and public service investments concentrate predominantly in the original urban cores, leaving newly merged districts at a relative disadvantage [37]. In addition, the fiscal management system [38] faces considerable pressure, as merged entities are required to integrate budgets and harmonize spending priorities, which can initially lead to disparities in service provision.

Although prior research has yielded valuable insights into the determinants of policy satisfaction and the effects of the TMP, several critical aspects warrant further exploration.

First, the majority of existing studies adopt a macro-level perspective, leaving a pronounced gap in the analysis of micro-level individual policy satisfaction and its underlying drivers. Second, while most research emphasizes the economic and social impacts of the township-town merger policy, insufficient attention has been devoted to its role in cultural integration and social security within the merged areas. Although these factors may not directly translate into personal economic gains, they constitute a fundamental component of the social environment, thereby exerting a significant influence on residents' policy satisfaction. Third, some previous research on boundary changes compared different regions to show differences in policy satisfaction. However, these regions have markedly different developmental contexts before boundaries change. Their geographic, economic, and cultural disparities confound the assessment of policy effects, thereby limiting the validity of conclusions about the merger policy's pure impact. Addressing these micro-level and comparative gaps is vital for informing the design of administrative restructuring policies that are both equitable and effective.

## Materials and methods

### Overview of the study area

The study area is located in Jiangning District of Nanjing City, occupying the southwestern part of the district along the southern bank of the lower Yangtze River. As shown in Fig 1, Lukou Subdistrict and Hengxi Subdistrict are geographically contiguous and comparable in size.

In 2000, Jiangning, formerly a county, underwent the administrative transition from county to district, formally becoming part of Nanjing City. This transformation required a shift from a traditional rural administrative model to an urban district governance structure. The implementation of the township-town merger policy in Jiangning District facilitated the consolidation of fragmented administrative resources and the establishment of an administrative framework aligned with Nanjing's broader urban development objectives. Building on this foundation, this study focuses on Jiangning District, with particular emphasis on comparing Lukou and Hengxi Subdistricts at the micro level to analyze the underlying determinants of residents' satisfaction with TMP and the disparities in policy satisfaction between the two Subdistricts.

These two Subdistricts exhibit the following two characteristics. (1) Lukou and Hengxi Subdistricts share comparable policy paths. Within the same township-town merger policy framework implemented in Jiangning District in 2006, both Subdistricts underwent administrative restructuring. The former Lukou town was reconstituted as Lukou Subdistrict, whereas Taowu town was merged with Hengxi town and Danyang town to form the new Hengxi Subdistrict. At the time of the merger, the two areas exhibited similar political, economic, and cultural characteristics. Following the implementation of the TMP, both attained equivalent administrative status as Subdistricts within Jiangning District. Furthermore, the timing and nature of their administrative adjustments were identical, providing a consistent institutional context and spatial framework for comparative analysis.

(2) Lukou and Hengxi Subdistricts diverged in their development approaches after the TMP. Despite undergoing the same policy intervention, the two regions pursued distinct paths shaped by their resource endowments and functional

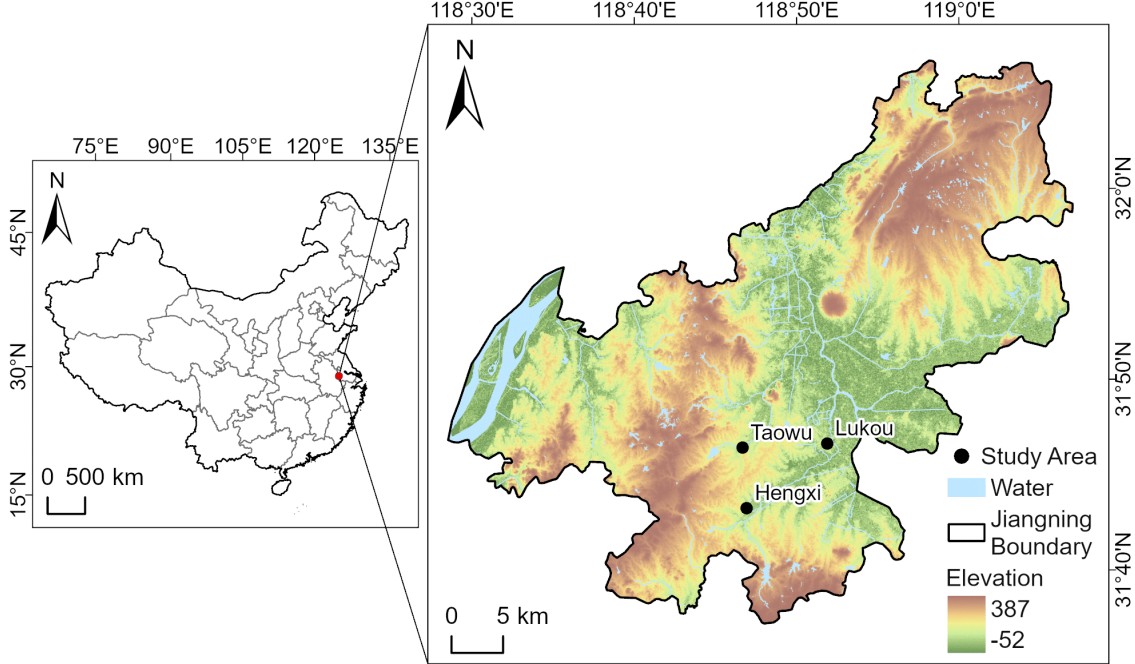

**Fig 1. Location of Lukou and Hengxi subdistricts.**

---

 

positioning. Leveraging the strategic advantage of Lukou International Airport, Lukou Subdistrict became deeply integrated into the airport economic zone, experiencing rapid expansion in export-oriented industries and accelerated urbanization. In contrast, although similarly proximate to the airport, Hengxi Subdistrict emphasized supporting industries and modern agriculture, resulting in a transformation pace and development model that markedly differed from Lukou's. These contrasting conditions justify the selection of Lukou and Hengxi Subdistricts as analytical units, and controlling for their initial economic and cultural similarities strengthens the comparability of residents' satisfaction with the TMP. On the other hand, the divergence in development models following policy implementation provides an appropriate context for investigating the underlying drivers of policy satisfaction.

## Questionnaire design and data collection

Data for this study were obtained through a structured survey designed to evaluate residents' satisfaction with the TMP in Lukou and Hengxi Subdistricts. The survey employed a mixed-mode approach, combining both online and offline administration to ensure broad coverage and representativeness. Online data collection started on September 28th, 2024, and ended on March 16th, 2025. Offline data collection was conducted on three occasions—September 28th, 2024, January 11th, 2025, and February 20th, 2025. The survey sites included Lukou Subdistrict Office, Maoting Community, Hengxi Subdistrict Office, Taowu Cinema Square, and Taowu Community. These sites were deliberately selected as they represent key public gathering areas where residents frequently interact.

Within each site, respondents were randomly selected to ensure representativeness. Respondents were eligible if they had resided in the subdistrict for at least 5 years to ensure familiarity with the TMP's long-term effects and were 18 years or older. 20 face-to-face interviews were analyzed using thematic analysis. Interview recordings were transcribed verbatim into text by apps. Researchers independently coded the text to identify key themes concerning identity recognition, expectancy formation, and social comparison.

After data screening and validation, a total of 243 valid questionnaires were retained for subsequent analysis. Among the respondents, 128 were male (52.67%) and 115 were female (47.33%). Regarding age distribution, 28 respondents were aged 18–30 (11.52%), 71 were aged 31–45 (29.22%), 96 were aged 46–60 (39.51%), and 48 were over 60 years old (19.75%). In terms of occupation, 62 were enterprise employees (25.51%), 30 were farmers (12.35%), 16 were government staff (6.58%), 14 were teachers (5.76%), 85 were freelancers (34.98%), and 36 were retirees (14.81%).

In addition to capturing respondents' individual characteristics, the questionnaire includes six primary dimensions: population development, infrastructure, environmental improvement, economic growth, social security, and cultural development. Specifically, the infrastructure dimension consists of four indicators. They are transportation networks, water conservancy facilities, communication broadband, and environmental protection infrastructure. The social security dimension comprises recreational activities, medical and healthcare services, sports facilities, and related aspects. The cultural development dimension is measured through three indicators: traditional festivals, cultural heritage preservation, and tourism development. A detailed description of the indicator system is provided in S1 Table.

Most items in the questionnaire were designed as scaled questions, measured on a five-point Likert scale, in which respondents were asked to indicate the extent of their agreement with given statements. For instance, in response to the question, "Has the township-town merger policy promoted development in this aspect?", a score of 1 represented "strongly disagree," while a score of 5 denoted "strongly agree." In addition, the questionnaire gathered respondents' overall perceptions of the township-town merger policy and their level of agreement regarding its contribution to local development. Reliability and validity tests were performed on all scaled items, with detailed results presented in Table 1. The overall Cronbach's alpha coefficient reached 0.940, and the reliability of each major dimension exceeded 0.75, indicating strong internal consistency. Furthermore, all KMO values were above 0.5, and Bartlett's tests of sphericity were statistically significant, confirming the questionnaire's robust construct validity.

Table 1. Reliability and Validity Analysis of Questionnaire Scale items.

| Questionnaire Items | Reliability | Validity | |
|---|---|---|---|
| | | KMO | Bartlett |
| All items | 0.940 | | |
| Population Development Effect | 0.630 | 0.598 | 97.652*** |
| Infrastructure Improvement Effect | 0.790 | 0.686 | 219.231*** |
| Environmental Improvement Effect | 0.830 | 0.751 | 378.647*** |
| Economic Growth Effect | 0.788 | 0.500 | 131.825*** |
| Social Security Effect | 0.818 | 0.758 | 332.420*** |
| Cultural Development Effect | 0.790 | 0.695 | 220.105** |

Note: Reliability test was conducted using Cronbach's Alpha, and validity analysis was performed using the KMO and Bartlett's sphericity test. Significance levels: * $p < 0.1$, ** $p < 0.05$, *** $p < 0.01$.

## Ethical consideration

This study does not require formal ethical approval, as it aligns with the general conditions for exemption from ethical review as commonly adopted in both national and international practices and can therefore be exempted from ethical review.

1.  It constitutes an anonymous questionnaire survey within the field of social sciences. The questionnaire does not record identifiable information such as name, ID number, or residential address. It does not involve biomedical experiments, the collection of sensitive personal information, the imposition of psychological stress, or any form of interventional procedures.

2.  Before conducting the research, we attached the informed consent forms to both the online and offline questionnaires. Participants must explicitly check the option stating "I have read and understood the above information and voluntarily agree to participate in this survey" before proceeding to the questionnaire section. This process ensures that participation is entirely voluntary and informed.

3.  This study entails minimal anticipated risk. It does not involve minors, will not subject participants to any form of deception or physical/psychological harm, and implements thorough anonymization and confidentiality measures, resulting in an extremely low risk of privacy leakage.

## Model selection

First, since the dependent variable—policy satisfaction—is binary, this study employs a nonlinear Logit model to examine the determinants of residents' satisfaction with the TMP from the perspectives of both individual characteristics and perceived policy effects. Second, to further explore the sources of satisfaction disparities between residents of Hengxi and Lukou Subdistricts, the Fairlie decomposition method is applied to decompose the intergroup differences and quantify the relative contribution of each influencing factor.

(1) Determinants of influencing satisfaction: Logit model

A binary logit discrete choice model is employed to estimate the determinants influencing residents' satisfaction with the TMP. The model is specified as follows:

$$\Pr\left(y_i = j | X\right) = F(X\beta) = \frac{\exp(X'\beta)}{1 + \exp(X'\beta)}$$

(1)

Here, $j = 1$ denotes that residents express a relatively high level of agreement with the TMP, while $j = 0$ signifies a relatively low level of agreement. $X$ represents the independent variables influencing residents' policy satisfaction, encompassing both individual characteristics and perceived policy effects. $F$ denotes the logistic function. Unlike linear models, $\beta$ does not represent a direct marginal effect but rather the marginal change in the log-odds ratio. Considering the potential interdependence among residents within the same Subdistrict, the standard errors are clustered at the Subdistrict level.

(2) Intergroup difference decomposition model

Since this study employs a nonlinear binary Logit discrete choice model with a dependent variable taking binary values of 0 or 1, the traditional Oaxaca-Blinder decomposition method is inapplicable. Consequently, we adopt the nonlinear Oaxaca-Blinder approach [39] to decompose the intergroup differences analyzed herein. This nonlinear model partitions the observed intergroup disparities into explainable and unexplainable components. The specific decomposition model is specified as follows:

$$E(y_l) - E(y_t) = \left[\sum_{i=1}^{n^l} \frac{F\left(X_i^l \beta^t\right)}{n^l} - \sum_{i=1}^{n^t} \frac{F\left(X_i^t \beta^t\right)}{n^t}\right] + \left[\sum_{i=1}^{n^l} \frac{F\left(X_i^l \beta^l\right)}{n^l} - \sum_{i=1}^{n^l} \frac{F\left(X_i^l \beta^t\right)}{n^l}\right] \tag{2}$$

Equation (2) uses the estimated coefficients of Hengxi Subdistrict as the benchmark to decompose the mean difference in satisfaction with the township-town merger policy between Lukou ($l$) and Hengxi ($t$) into two components. Here, $n^l$ and $n^t$ represent the sample sizes of surveyed residents in the two areas, respectively. The first term on the right side is the explained part, which arises from differences in the explanatory variables $X$ between the two regions. The second term on the right side is the unexplained part, resulting from differences in the coefficients of the explanatory variables.

To mitigate the potential influence of benchmark selection, this study follows Fairlie's approach by using the estimated coefficients derived from the pooled sample of residents from both Subdistricts as $\beta^t$ in the first term on the right-hand side. Moreover, within the explained component, the contribution of any specific variable, such as $X_1$, to the overall difference can be expressed as:

$$\sum_{i=1}^{n^l} \frac{F\left(X_{1i}^l \beta_1^t + other_i^l \beta_{other}^t\right)}{n^l} - \sum_{i=1}^{n^t} \frac{F\left(X_{1i}^t \beta_1^t + other_i^t \beta_{other}^t\right)}{n^t} \tag{3}$$

**Variables**

The dependent variable is a binary indicator measuring residents' satisfaction with the policy, derived from the survey question: "Do you believe the township-town merger policy has significantly promoted local development?" Responses scored between 1 and 3 were coded as 0, representing lower satisfaction, while scores of 4 or 5 were coded as 1, representing higher satisfaction. In our study, the operationalization of the dependent variable is based solely on a single global question regarding whether the policy encourages local development. It is because a single-item measure directly captures satisfaction as an outcome while avoiding the subjective bias introduced by weighting multidimensional indicators. Therefore, following Fiorio & Florio [40], we measure residents' policy satisfaction using a single-item measure.

Moreover, in this study, we controlled for individual characteristics to mitigate the interference of confounding factors on the conclusions to some extent. Individual characteristics considered in the study include gender, age, occupation, and cognitive ability. Gender is coded as a binary variable, with males assigned a value of 1 and females a value of 0. Age is categorized into four groups: 18–30 years, 31–45 years, 46–60 years, and over 60 years, with the oldest group serving

as the reference category; three dummy variables represent the remaining age groups. Occupation is classified into six categories: enterprise employee, farmer, government staff, teacher, self-employed, and retiree. Retirees are used as the reference group, and five dummy variables are constructed to capture the other occupational categories. Considering that cognitive abilities and the capacity to adapt to new things gradually decline with age among the elderly, we use the logarithm of the time (in seconds) spent on completing the questionnaire as a proxy variable for measuring the cognitive abilities of the elderly. The longer the time taken to complete the questionnaire, the lower their cognitive abilities.

To statistically assess common method bias, we conducted Harman's one-factor test [41,42]. Specifically, as shown in the exploratory factor analysis results in Table A2, the first principal component accounts for 26.72% of the total variance. This value is substantially lower than the conventional threshold of 50% [43], suggesting that common method bias is not a serious concern in this study.

To capture the policy effects shaping residents' satisfaction, six dimensions are assessed: population development, infrastructure improvement, environmental enhancement, economic growth, social security, and cultural development. Composite indices for each dimension were constructed by assigning weights to sub-indicators using the entropy method. Table 2 presents descriptive statistics for Lukou and Hengxi Subdistricts, highlighting the differences between the two groups. Overall, notable disparities are evident in both residents' policy satisfaction levels and across all assessed policy effect dimensions.

## Results

### Analysis of the determinants of policy satisfaction

Table 3 reports the results of the baseline regression and the marginal effects of each dependent variable at the sample mean. Column (1) shows the relationship between policy effects and residents' policy satisfaction without controlling for individual characteristics. Column (2) shows the relationship between policy effects and residents' policy satisfaction when controlling for individual effects. With the inclusion of individual characteristics, the employment incentive effect, social security effect, and cultural development effect all show a positive correlation with policy satisfaction at the 5% significance level. Column (3) reports the ordinary standard errors, and Column (4) reports the standard errors clustered at the regional and temporal levels. The results still indicate that the employment incentive effect (coefficient = 1.3958, $p < 0.01$), social security effect (coefficient = 2.8941, $p < 0.01$), and cultural development effect (coefficient = 2.8488, $p < 0.05$) all exhibit a significant positive correlation with policy satisfaction. Notably, the social security effect represents the most influential predictor, followed by the cultural development effect, highlighting the substantial roles of cultural preservation and social safety nets in shaping policy approval.

Column (5) reports the marginal effects of each variable at the sample mean. Specifically, holding other variables constant, for each one-unit increase in the social security effect, the probability that residents remain satisfied with the TMP policy increases by 70.5%; for each one-unit increase in the cultural protection effect, the probability that residents remain satisfied with the TMP policy increases by 69.4%; and for each one-unit increase in the employment incentive effect, the probability that residents remain satisfied with the TMP policy increases by 34%.

To further validate the robustness of these findings, a series of robustness checks was conducted. First, we use an alternative dependent variable as a robustness check. Specifically, we measure policy satisfaction on a 1–5 scale, where higher scores indicate greater satisfaction. In terms of model specification, because the dependent variable is an ordered discrete value, we re-estimate the regression using an ordered logit model. The results are presented in Column (1) of S2 Table. The result reveals that the employment incentive effect, social security effect, and cultural development effect still have a positive impact on residents' policy satisfaction at the 5% significance level, confirming the appropriateness of the variable specification. Second, we included month fixed effects in the model. These fixed effects help rule out the influence of time-specific events on the policy evaluations of older adults, while also mitigating sample selection bias arising from different survey periods. The results are shown in Column (2) of Table A1, and the baseline conclusions remain robust.

**Table 2. Description of variables in the model (N = 243).**

| Variables | | Total | Lukou | Hengxi | Mean Difference |
|---|---|---|---|---|---|
| Dependent Variable | Policy Satisfaction | 0.444 | 0.561 | 0.385 | 0.175*** |
| | | (0.498) | (0.499) | (0.488) | [0.067] |
| Personal Characteristics | Gender | 0.527 | 0.5 | 0.54 | −0.04 |
| | | (0.500) | (0.503) | (0.5) | [0.068] |
| | Age 18 | 0.115 | 0.183 | 0.081 | 0.102** |
| | | (0.320) | (0.389) | (0.273) | [0.043] |
| | Age 31 | 0.292 | 0.183 | 0.348 | −0.165*** |
| | | (0.456) | (0.389) | (0.478) | [0.061] |
| | Age 46 | 0.395 | 0.512 | 0.335 | 0.177*** |
| | | (0.490) | (0.503) | (0.474) | [0.065] |
| | Occupation Agriculture | 0.123 | 0.098 | 0.137 | −0.039 |
| | | (0.330) | (0.299) | (0.345) | [0.045] |
| | Occupation Firm | 0.255 | 0.317 | 0.224 | 0.093 |
| | | (0.437) | (0.468) | (0.418) | [0.059] |
| | Occupation Government | 0.066 | 0.085 | 0.056 | 0.029 |
| | | (0.249) | (0.281) | (0.23) | [0.034] |
| | Occupation Teaching | 0.058 | 0.061 | 0.056 | 0.005 |
| | | (0.233) | (0.241) | (0.23) | [0.032] |
| | Occupation Free | 0.350 | 0.305 | 0.373 | −0.068 |
| | | (0.478) | (0.463) | (0.485) | [0.069] |
| | Used Time | 5.739 | 5.569 | 5.827 | −0.257** |
| | | (0.052) | (0.076) | (0.068) | [0.110] |
| Policy Effect | Population Development | 0.623 | 0.712 | 0.578 | 0.135*** |
| | | (0.230) | (0.220) | (0.222) | [0.030] |
| | Infrastructure Improvement | 0.641 | 0.682 | 0.620 | 0.062* |
| | | (0.248) | (0.250) | (0.246) | [0.034] |
| | Environmental Improvement | 0.656 | 0.722 | 0.622 | 0.099*** |
| | | (0.232) | (0.212) | (0.235) | [0.031] |
| | Economic Growth | 0.580 | 0.675 | 0.532 | 0.144*** |
| | | (0.253) | (0.214) | (0.257) | [0.033] |
| | Social Security | 0.611 | 0.707 | 0.562 | 0.144*** |
| | | (0.015) | (0.025) | (0.019) | [0.032] |
| | Cultural Development | 0.524 | 0.552 | 0.510 | 0.042 |
| | | (0.252) | (0.242) | (0.257) | [0.034] |
| N | | 243 | 82 | 161 | |

Note: The values in parentheses are standard deviations, the values in square brackets are the standard errors of the t-test for the difference between the two group means, and * p < 0.1, ** p < 0.05, *** p < 0.01.

## Disparities in policy satisfaction between residents of the two subdistricts

(1) Comparison of Policy Satisfaction Between Hengxi and Lukou Subdistricts

According to Table 4, residents of Lukou Subdistrict have significantly higher policy satisfaction (coefficient = 0.561) compared to those in Hengxi Subdistrict (coefficient = 0.385), resulting in a satisfaction gap of 0.176 between the two areas.

**Table 3. Overall description of policy satisfaction.**

| | (1) Policy Satisfaction | (2) Policy Satisfaction | (3) Policy Satisfaction | (4) Policy Satisfaction | (5) Margin Effects |
|---|---|---|---|---|---|
| Population Development Effect | 1.4119** | 1.0650 | 1.0650 | 1.0650 | 0.259 |
| | (0.7184) | (0.8544) | (1.0078) | (1.2489) | (0.211) |
| Infrastructure Improvement Effect | 0.4057 | 0.3068 | 0.3068 | 0.3068 | 0.075 |
| | (2.0788) | (2.3137) | (1.0795) | (1.3809) | (0.562) |
| Environmental Improvement Effect | 0.2285 | 0.6916 | 0.6916 | 0.6916 | 0.168 |
| | (3.5689) | (3.2677) | (1.2713) | (1.4542) | (0.797) |
| Income Growth Effect | −0.6951 | −0.9425 | −0.9425 | −0.9425 | −0.229 |
| | (1.2467) | (1.2393) | (0.7629) | (0.9470) | (0.304) |
| Employment Incentive Effect | 1.4578*** | 1.3958** | 1.3958* | 1.3958*** | 0.340*** |
| | (0.5100) | (0.5538) | (0.7662) | (0.4712) | (0.131) |
| Social Security Effect | 2.2970*** | 2.8941*** | 2.8941** | 2.8941*** | 0.705*** |
| | (0.3480) | (0.0870) | (1.2555) | (1.0764) | (0.013) |
| Cultural Development Effect | 3.1524** | 2.8488** | 2.8488*** | 2.8488*** | 0.694** |
| | (1.2752) | (1.1858) | (0.8574) | (0.6170) | (0.281) |
| Gender | | 0.1714 | 0.1714 | 0.1714 | 0.042 |
| | | (0.4103) | (0.3454) | (0.3117) | (0.099) |
| Age_18 | | 0.6904 | 0.6904 | 0.6904 | 0.168 |
| | | (0.8645) | (0.7404) | (0.8243) | (0.209) |
| Age_31 | | 0.5499* | 0.5499 | 0.5499 | 0.134* |
| | | (0.3179) | (0.6105) | (0.6745) | (0.076) |
| Age_46 | | 0.6235** | 0.6235 | 0.6235 | 0.152** |
| | | (0.2798) | (0.5281) | (0.5376) | (0.070) |
| Occupation_Agriculture | | −0.3812 | −0.3812 | −0.3812 | −0.093 |
| | | (0.5998) | (0.6477) | (0.9345) | (0.147) |
| Occupation_Firm | | 0.1475*** | 0.1475 | 0.1475 | 0.036*** |
| | | (0.0154) | (0.6106) | (0.7250) | (0.003) |
| Occupation_Government | | 0.5716 | 0.5716 | 0.5716 | 0.139 |
| | | (0.3813) | (0.8432) | (0.8372) | (0.091) |
| Occupation_Teaching | | −1.2339 | −1.2339 | −1.2339 | −0.300 |
| | | (0.8246) | (0.9029) | (1.3728) | (0.197) |
| Occupation_Free | | −0.2226 | −0.2226 | −0.2226 | −0.054 |
| | | (0.4515) | (0.5881) | (0.7847) | (0.109) |
| ln_time | | 0.1699* | 0.1699 | 0.1699 | 0.041** |
| | | (0.0883) | (0.2229) | (0.3068) | (0.021) |
| _cons | −5.0998*** | −6.5936*** | −6.5936*** | −6.5936*** | |
| | (0.0883) | (1.2857) | (1.6776) | (1.4926) | |
| N | 243 | 243 | 243 | 243 | |
| pseudo R² | 0.285 | 0.306 | 0.306 | 0.306 | |

Note:* p<0.1, ** p<0.05, *** p<0.01. Column 1 and Column 2 use cluster-level regional mislabeling, Column 3 uses standard error, and Column 4 uses a dual standard error for both region and time.

**Table 4. Analysis of intergroup differences.**

| | Coefficient | Degree of Explanation |
|---|---|---|
| Policy Satisfaction – Lukou subdistrict | 0.561 | |
| Policy Satisfaction – Hengxi subdistrict | 0.385 | |
| Total Difference | 0.176 | |
| Individual Characteristics | 0.016 | 0.090 |
| Policy Effect | 0.159 | 0.904 |
| Unexplained Portion | 0.001 | 0.006 |

To unpack the drivers of this gap, we employed a Fairlie decomposition analysis. The results indicate that 99.4% of the observed difference in policy satisfaction can be explained by the set of covariates included in our model, leaving only a negligible 0.6% attributable to unobserved factors. Regarding the composition of explanatory factors, policy effects stand out as the primary driver of the satisfaction gap between the two Subdistricts, accounting for 90.4% of the difference, which far exceeds the contribution of individual characteristics at 9%. This finding indicates that the substantial disparities in policy satisfaction between Lukou and Hengxi stem from differences in policy implementation quality, execution approaches, and the direct benefits achieved, rather than from individual characteristics.

Table 5 presents the results of the Fairlie decomposition analysis, which further disaggregates the policy-driven component of the satisfaction gap between the two subdistricts into specific policy effect dimensions. Among all policy-related factors, the social security effect is the dominant contributor to the observed disparity. With a contribution value of 0.0707 ($p < 0.01$), it alone accounts for 40.20% of the total satisfaction gap. The employment incentive effect is the second most impactful factor since its contribution value of 0.0372 ($p < 0.01$) explains 21.20% of the gap. Additionally, the population development effect (contribution = 0.0258) and cultural development effect (coefficient = 0.0209, $p < 0.01$) contribute 14.67% and 11.88% of the total difference, respectively.

Collectively, the three core policy effects, social security, employment incentive, and cultural development, explain 73.28% of the observed difference in policy satisfaction. This is to say that they are the key drivers of inter-subdistrict satisfaction disparities.

(2) Comparison of policy satisfaction among Taowu town, Hengxi town and Lukou town

Prior to the implementation of the township-town merger policy, Taowu, Hengxi, and Lukou were all towns in Jiangning, sharing the same administrative status. As mentioned in "Overview of the study area ", following the merger, Taowu town was incorporated into Hengxi town, and subsequently the two merged together to form the new Hengxi Subdistrict. Taowu town was dissolved after TMP. Therefore, this study further analyzes whether Taowu, as a dissolved entity, exhibits

**Table 5. Fairlie decomposition of intergroup differences.**

| | Contribution Value | Standard Error | Contribution Percentage |
|---|---|---|---|
| Population Development Effect | 0.0258 | 0.0236 | 0.1467 |
| Infrastructure Effect | 0.0032 | 0.0253 | 0.0182 |
| Environmental Improvement Effect | 0.0113 | 0.0567 | 0.0642 |
| Income Growth Effect | −0.0101 | 0.0165 | −0.0574 |
| Employment Incentive Effect | 0.0372*** | 0.0138 | 0.212 |
| Social Security Effect | 0.0707*** | 0.0090 | 0.4020 |
| Cultural Development Effect | 0.0209*** | 0.0080 | 0.1188 |

Note:* $p < 0.1$, ** $p < 0.05$, *** $p < 0.01$.

differences in policy satisfaction compared to Lukou and Hengxi, and explores the underlying reasons for observed disparities in residents' satisfaction in the three regions.

As shown in Table 6, a substantial gap in policy satisfaction exists between residents of Taowu and Lukou Subdistricts. Specifically, the mean policy satisfaction score for Lukou residents is 0.5609, which is significantly higher than the score of 0.4347 for Taowu residents, yielding a total difference of 0.1262.

To identify the key drivers of this gap, we conducted a Fairlie decomposition analysis. The results reveal that the population development effect is the most influential factor, with a coefficient of 0.1085 ($p < 0.01$), accounting for 85.97% of the total difference. This is followed by the social security effect (coefficient = 0.0624, $p < 0.01$), and the employment incentive effect (coefficient = 0.0393, $p < 0.01$), accounting for 49.45% and 31.14% of the total difference, respectively.

These findings indicate that the higher policy satisfaction among Lukou residents is primarily driven by their stronger perceptions of policy effectiveness in the areas of population development, employment incentives, and social security.

Table 7 presents the policy satisfaction differences between Taowu town and the original Hengxi town before TMP. The mean policy satisfaction of Taowu residents is 0.4347, and that of Hengxi residents is 0.3652, with a total difference of 0.0695, indicating that Taowu residents report higher policy satisfaction than those in Hengxi. The environmental improvement effect is the primary positive contributor (coefficient = −0.0277, $p < 0.01$), accounting for the 39.86% of the total difference. This is followed by the cultural development effect (coefficient = 0.0097). In contrast, the employment incentive effect (coefficient = 0.0224, $p < 0.01$) and social security effect (coefficient = 0.0229) are the main negative contributors.

Furthermore, we re-examine the Fairlie decomposition using a probit model, with results presented in S4–S6 Tables. The results show that no substantial changes in significance or coefficient direction, and the fluctuation in the contribution ratio did not exceed 5%.

## Discussion

The results offer critical insights into the determinants of residents' policy satisfaction and the substantial disparities between Lukou and Hengxi subdistricts. These differences, in turn, shape residents' perceptions and evaluations.

### Discussion on influencing factors

In comparison with demographic characteristics, policy implementation outcomes are the core drivers of residents' evaluation (Table 3). The significant contributors to the policy satisfaction of the two subdistricts are the social security effect, the cultural development effect, and the employment incentive effect. This finding is consistent with the core view of public service satisfaction theory that actual policy benefits are the direct determinant of public evaluation [25].

**Table 6. Fairlie decomposition of policy satisfaction differences between Taowu and Lukou.**

|  | Coefficient | Contribution Percentage |
|---|---|---|
| Policy Satisfaction – Lukou | 0.5609 |  |
| Policy Satisfaction – Taowu | 0.4347 |  |
| Total Difference | 0.1262 |  |
| Population Development Effect | 0.1085*** | 0.8597 |
| Infrastructure Effect | −0.0147 | −0.1165 |
| Environmental Improvement Effect | 0.0744 | 0.5895 |
| Income Promotion Level | −0.0454 | −0.3598 |
| Employment Incentive Effect | 0.0393*** | 0.3114 |
| Social Security Effect | 0.0624*** | 0.4945 |
| Cultural Development Effect | 0.0081 | 0.0642 |

Note:* $p < 0.1$, ** $p < 0.05$, *** $p < 0.01$.

**Table 7. Policy satisfaction differences between Taowu and Hengxi.**

|  | Coefficient | Contribution Percentage |
|---|---|---|
| Policy Satisfaction – Hengxi | 0.3652 |  |
| Policy Satisfaction – Taowu | 0.4347 |  |
| Total Difference | −0.0695 |  |
| Population Development Effect | 0.0007 | −0.0101 |
| Infrastructure Effect | −0.0059 | 0.0849 |
| Environmental Improvement Effect | −0.0277** | 0.3986 |
| Income Growth Effect | 0.0044 | −0.0633 |
| Employment Incentive Effect | 0.0224*** | −0.3223 |
| Social Security Effect | 0.0229 | −0.3295 |
| Cultural Development Effect | −0.0097 | 0.1396 |

Note:* $p < 0.1$, ** $p < 0.05$, *** $p < 0.01$. Hengxi refers to Hengxi town before the implementation of TMP.

As the most influential contributor, social security resonates closely with Marshall's theory of citizenship, which proposes that the provision of social security constitutes a foundational pillar of governmental legitimacy and citizen satisfaction [44]. Enhancements in the social safety net, including improved access to healthcare and pensions, deliver tangible benefits to residents, alleviate concerns about future uncertainties, and foster a pronounced sense of security, which in turn translates directly into elevated levels of policy approval. The second influential contributor is the cultural development effect. Cultural initiatives such as the promotion of traditional arts and community cultural events not only enrich residents' spiritual lives but also strengthen social cohesion and a sense of belonging. For example, the local government's investment in the "Lukou Fur Town" project, which celebrates over 1500 years of regional fur craftsmanship, wins residents' approval. The high rankings of the social security effect and the cultural development effect indicate that non-economic factors play a pivotal role in shaping public perception of policy effectiveness.

The coefficients for the employment incentive effect are significantly positive at the 10% level, indicating that residents perceive the policy as successful when it generates local employment opportunities. Access to jobs within one's hometown is closely linked to broader local economic development, highlighting that the core economic objectives of the township-town merger policy are important for public approval.

## The mechanism of regional policy satisfaction difference

**Differences in development models across regions.** The significant policy satisfaction gap between Lukou and Hengxi Subdistricts (Table 5) is mainly driven by policy effect variables accounting for 90.4% of the difference. The most influential factors, in order, are the social security effect, the employment incentive effect, and the cultural development effect. This finding aligns with insights obtained from interviews with local residents. For instance, when asked about their overall perceptions of the TMP, some old respondents in Lukou Subdistrict expressed high satisfaction with the policy. This is because every elderly person over 70 in Lukou receives a special subsidy of CNY 180 per year, in addition to a monthly pension of CNY1000. Others noted that Lukou experienced rapid economic development after the TMP and quickly became a leader in Jiangning District.

*Q: What do you think about the township-town merger policy?*

*A1: Well, to be honest... I don't really understand all those policies. But I can tell you...since this TMP thing, Lukou has gotten a lot better. You see, the government gives us old folks money every year. It's 180 kuai... that's very good. And you know, that money can buy my food for a whole week.*

*A2: We ordinary folks just go about our daily lives. But over the past 20 years or so, the economy here in Lukou has really taken off...life is much better than it used to be.*

The reason why Lukou Subdistrict can offer better social security to residents than Hengxi Subdistrict is their differentiated development paths after the implementation of TMP. As mentioned in the research area section, Lukou Subdistrict has leveraged its proximity to the international airport to develop airport-oriented industries. To be more specific, Lukou has currently focused on the high-end intelligent equipment, auto parts and Nanjing International Distribution and Processing Center (Phase I), forming the rudiment of a 10-billion-level industrial cluster.

While vigorously developing its economy, Lukou Subdistrict has simultaneously increased its investment in social security and cultural preservation. On one hand, the robust fiscal base has provided a great number of job positions, enabled greater investment in social security and the preservation of cultural heritage, such as the Shuijingshu Village Horse Lantern performance, which has strengthened residents' economic security and cultural belonging. On the other hand, these investments in social security and cultural preservation have elevated resident satisfaction with the TMP. This dual focus has created a mutually reinforcing, win-win situation.

In contrast, Hengxi Subdistrict's attempt to replicate Lukou's strategy achieved only partial success, leaving it reliant on residual resources and fiscally constrained. Consequently, Hengxi has been unable to invest comparably in social programs or cultural preservation, with historical assets like Shuyun Bridge and a century-old street remaining underutilized. The annual Hengxi Watermelon Festival, lacking cultural depth, fails to foster a strong local identity.

**Psychological mechanisms.** In the comparison between Taowu and Lukou, there is a substantial gap in policy satisfaction (Table 6). This indicates that Taowu underperforms compared to Lukou in overall economic aspects, especially in population development, social security, and employment incentives. On one hand, these findings are consistent with Lai [12], who discovered that township mergers led to notable population agglomeration, which ultimately enhances residents' policy satisfaction. On the other hand, the interviews also reveal that in Lukou, not only is the population more concentrated, but the area has also attracted numerous migrants who are satisfied with the TMP. For instance, one interviewee stated that her son chose to settle and work in Lukou because of the city's well-organized job fairs and effective job-seeking WeChat public accounts, which means good job opportunities. Meanwhile, interviews with the residents of Taowu show that they are dissatisfied with the significant difference in the development process between the two places.

*Q: "Are you satisfied with the development of Taowu after the township merger?"*

*A: No, not at all. Before, Taowu was the top dog...Lukou and Hengxi couldn't even compare with us. But now look, Taowu is gone, and we can't even be considered the third anymore.*

From the interview findings, we can abstract a comparative mechanism: after being merged, Taowu lost its independent administrative status, leading to a weaker collective identity among its residents. As a result, they perceive their own development as lagging behind that of neighboring Lukou, giving rise to an upward social comparison. This perceived disparity, combined with Lukou's rapid growth, makes Taowu residents feel relatively dissatisfied with the TMP. Since Hengxi Subdistrict's development was considered inferior to that of Lukou, Taowu residents, now part of Hengxi, express weaker collective identification with their new administrative unit. This combination of low identity recognition and unfavorable social comparison ultimately contributes to Taowu's lower satisfaction. As previous studies have consistently demonstrated that upward social comparison tends to reduce residents' satisfaction and sense of recognition [45,46], particularly in the context of administrative adjustment and resettlement programs [46], our study further extends these findings to the scenario of township merger and regional restructuring.

Interestingly, residents of the former Hengxi town reported lower satisfaction with the TMP (Table 7), despite the fact that the two towns have now been merged into one subdistrict. In more detail, Taowu residents express higher policy

satisfaction than those in Hengxi. A possible explanation lies in the relatively strong sense of identity among Hengxi residents, whose policy satisfaction is shaped primarily by temporal comparisons with their own past, particularly in the domains of environment and culture (Table 7). Although Hengxi outperforms Taowu in terms of employment incentives, its shortcomings in environmental and cultural outcomes contribute to its lower overall policy satisfaction. This pattern is consistent with our interview findings.

> Q: "Are you satisfied with the development of Hengxi after the township merger?"

> A: It's okay, I guess. The biggest thing is that Hengxi just isn't as lively as it used to be. We used to have opera troupes performing, and the March 17th fair would be packed with people...now that's all gone. The old street was such a wonderful place, but it's all run-down now.

In summary, residents in areas with weaker collective identity tend to compare themselves both with their own past and with more developed neighboring regions. This is a pattern often associated with lower levels of policy satisfaction. Meanwhile, residents with a stronger collective identity would like to compare themselves with their own past.

## Conclusions and implications

### Conclusions

This study obtains some meaningful findings on differences in resident satisfaction and driving factors of the TMP through analyzing 243 surveys collected in Lukou and Hengxi Subdistricts.

Firstly, compared to individual characteristics, policy effects prove to be the main drivers of resident satisfaction. Key influencing factors include employment incentives, social security, and cultural development, all of which show significant positive impacts on policy satisfaction.

Secondly, a clear regional disparity exists among Lukou, Hengxi and Taowu. Satisfaction with TMP in Lukou Subdistrict substantially exceeds that in Hengxi Subdistrict. Policy effects account for 90.4%of this variation. Further comparison reveals that residents of Lukou and Hengxi exhibit a strong sense of collective identity with the post-merger Subdistricts, whereas Taowu, as the merged entity, shows a lower sense of identification. The low collective identification among Taowu residents leads to a weaker sense of belonging to the post-merger Subdistrict, coupled with disparities compared with Lukou, contributing to their lower policy satisfaction. In contrast, Lukou residents demonstrate higher policy satisfaction due to a well-established social security system and well-designed employment incentive policies, as well as good cultural preservation following the merger. In comparison, Hengxi residents report lower satisfaction owing to insufficiency in cultural preservation and environmental improvements.

### Theoretical implications

On a theoretical level, considering that the theoretical summary presented in this paper is based on China's political system, it cannot be used as a theoretical framework to guide boundary changes in countries with different political structures. However, for other regions of China and countries with similar national conditions, this study makes theoretical contributions in three aspects.

Firstly, this research enhances our understanding of the driving factors behind policy satisfaction and thereafter uncovers the influences exerted by the top-down administrative boundary changes. By comparing Lukou Subdistrict and Hengxi Subdistrict as case studies, this research provides a theoretical reference for analyzing administrative boundary adjustments in China and abroad from a micro policy effects perspective. Our findings suggest that top-down boundary changes are not just governance tools but also a process of "policies creating politics." They substantially redefine citizens' rights and interests by reconfiguring the population, economic, and social structures through force. In contrast to bottom-up

boundary changes, which emphasize citizens' wills, top-down boundary changes should pay special attention to residents' policy feedback loops upon implementation so as to enhance residents' support and satisfaction.

Secondly, traditional theories believe that personal characteristics, family income, understanding of policy content and values, fairness perception, and implementation efficiency are key driving factors of policy satisfaction. This study, however, argues that non-economic factors, such as social security and cultural protection, also play an important role in policy satisfaction. Direct economic impacts are only one aspect of the overall driving factors. The diversity of driving factors highlights the inherent complexity of citizens' evaluation of policies, undermining the validity of evaluation systems that rely on singular, homogeneous indicators. This finding closely aligns with public service satisfaction models that emphasize the diversity of citizens' expectations and evaluation standards for government services [47].

Thirdly, this study proposes that subject identity exerts an influence on the process of how policy satisfaction is formed. Although previous research has established the impact of "resident identity" on policy satisfaction, its focus has predominantly been on how post-merger shifts in residents' individual identities influence policy perceptions. In contrast, this study places collective identity at the center of its analysis, thereby deepening this line of inquiry.

Our findings indicate that residents evaluate policy satisfaction through the dual perspectives of collective identity and analogous comparison. Regarding identity, residents with a stronger collective identity after TMP are more likely to perceive policy effects, whereas those with a weaker collective identity tend to have specific expectations, and their satisfaction diminishes when policy outcomes fall short. This finding strongly resonates with the social identity theory [48], which believes that individuals derive self-esteem from positive evaluations of their in-group. This manifests as a heightened tendency to perceive group-related affairs favorably. In contrast, residents with a weak collective identity lack this psychological foundation. Their satisfaction is consequently more dependent on instrumental factors, specifically the alignment between policy performance and their personal expectations.

In terms of comparison methods, the merging towns primarily relied on temporal comparisons, assessing policy effects based on conditions before and after implementation. Conversely, the merged towns employed both temporal comparisons and analogous comparisons, evaluating policy effects by benchmarking developments in adjacent regions. This reveals the mechanism through which entity differences influence policy satisfaction, highlighting how the choice of reference group is itself shaped by identity.

## Practical implications

In practice, some successful experiences and concrete measures adopted by the two Subdistricts can offer useful insights for countries with different political systems during their boundary changes. For example, after the TMP, Lukou did not focus solely on economic efficiency, but consistently prioritized cultural preservation and the cultivation of community identity. As a result, Lukou achieved both economic growth and high resident satisfaction. This outcome echoes the observation by Kössler and Fessha [49], who highlighted the ongoing challenge in African and European countries of balancing improved resource allocation through mergers or splits with the need to preserve local identity during boundary changes.

Firstly, boundary adjustment policies should address both cultural preservation and social security. In countries with diverse ethnic or cultural groups, overlooking issues of identity or welfare during restructuring can result in opposition [49]. Policymakers should identify and protect distinctive local traditions, such as Lukou's fur craftsmanship and traditional customs. Supporting cultural development helps maintain important community spaces and traditions, strengthens social bonds, and fosters a shared sense of identity in merged areas.

Secondly, building collective identity is key to policy satisfaction after administrative mergers. Compared with Lukou, the lower satisfaction observed among Taowu residents, linked to a weaker collective identity, shows the importance of encouraging community participation and organizing cultural activities. These measures can help residents in newly merged areas develop a stronger sense of belonging. And then, foster a strong sense of satisfaction with boundary changes.

Thirdly, narrowing regional satisfaction gaps calls for balanced and locally sensitive governance in merged towns. In fast-urbanizing countries, this shift helps prevent overemphasis on economic growth at the expense of inclusivity. Our findings indicate that differences in satisfaction between Lukou and Hengxi result not only from uneven development, but also from social comparisons intensified by weakened collective identity. To address this, governments should focus investments on cultural, environmental, and economic growth in less developed or newly merged areas, ensuring policies benefit all communities more equally.

Finally, it is important to help residents who have lost their hometowns due to boundary changes develop a new sense of shared identity. This can be supported by involving all original communities in planning and decision-making, and by holding events that celebrate and blend different local cultures. Such efforts can strengthen cohesion and belonging in newly merged areas, and may serve as a useful reference for other countries facing similar changes.

## Limitations

Despite the insights obtained from this study, several methodological limitations need to be recognized. Endogeneity problems arising from the restricted design of questionnaire items and unobserved variables may impede strict causal inference in our study. Hence, when interpreting the model results, we placed more emphasis on the correlations among variables.

Second, the findings of this paper are based on a small-sample survey. Future research, therefore, could further explore the factors influencing satisfaction with boundary change policies through large-scale surveys.

Third, the study areas are two neighboring towns and villages, where geographical proximity serves as an important condition for the mechanisms of social comparison and identity formation. When the scope of the study is extended to the national or even global level, these mechanisms may evolve, which warrants further investigation in future research.

## Supporting information

**S1 Fig. Location of Lukou and Hengxi Subdistricts.**
(TIF)

**S1 Table. Policy effect evaluation indicators.**
(DOCX)

**S2 Table. Robustness test.**
(DOCX)

**S3 Table. Principal components.**
(DOCX)

**S4 Table. Fairlie decomposition intergroup differences.**
(DOCX)

**S5 Table. Fairlie decomposition Taowu vs. Lukou.**
(DOCX)

**S6 Table. Table 6. Policy satisfactionvTaowu vs. Hengxi.**
(DOCX)

**S7 File. Informed consent statement.**
(DOCX)

**S8 File. Request for waiver of ethical review.**
(PDF)

**S9 File. Code.**
(DO)

**S10 File. Revised data.**
(DTA)

## Acknowledgments

The authors would like to thank all the respondents to this study.

## Author contributions

**Conceptualization:** Qiong Wang.

**Data curation:** Jie Sun, Yajie Yu, Changlin Zhang, Pingfan Hao.

**Methodology:** Qiong Wang, Hao Zhao.

**Software:** Yajie Yu.

**Writing – original draft:** Qiong Wang.

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
