## [Decision Letter · Decision Letter 0]

19 Jan 2026

Dear Dr. Zhao,

Thank you for submitting your manuscript to PLOS ONE. After careful consideration, we feel that it has merit but does not fully meet PLOS ONE’s publication criteria as it currently stands. Therefore, we invite you to submit a revised version of the manuscript that addresses the points raised during the review process.

We look forward to receiving your revised manuscript.

Kind regards,

Tatchalerm Sudhipongpracha

Academic Editor

PLOS One

Journal Requirements:

2. Thank you for providing your underlying data as Supporting Information.

We note that the data set contains text or data that is not in English. Please note that PLOS is an English-language publisher, so we require data sets to be provided in English as well. Please upload an English-language version of your data set.

This will also allow us to determine if your data follows PLOS standards per our Data Availability policy here: https://journals.plos.org/plosone/s/data-availability

(1) You may seek permission from the original copyright holder of Figure 1 to publish the content specifically under the CC BY 4.0 license.

4. Please include captions for your Supporting Information files at the end of your manuscript, and update any in-text citations to match accordingly. Please see our Supporting Information guidelines for more information: http://journals.plos.org/plosone/s/supporting-information....

Reviewers' comments:

Reviewer's Responses to Questions

**Comments to the Author**

1. Is the manuscript technically sound, and do the data support the conclusions?

Reviewer #1: Yes

Reviewer #2: Partly

2. Has the statistical analysis been performed appropriately and rigorously?

Reviewer #1: Yes

Reviewer #2: Yes

3. Have the authors made all data underlying the findings in their manuscript fully available?

Reviewer #1: Yes

Reviewer #2: Yes

4. Is the manuscript presented in an intelligible fashion and written in standard English?

Reviewer #1: No

Reviewer #2: Yes

Reviewer #1: This paper presents a relevant and meaningful empirical study on public satisfaction with the Township–Town Merger Policy (TMP). Based on a field survey of 243 respondents in two subdistricts (Lukou and Hengxi), the study makes novel contributions to the literature on regional administration and public governance, particularly through its analysis of intergenerational and interregional differences. There is no indication that the findings have been previously published, and the data used are primary and collected directly by the author.

From a methodological perspective, this study generally meets good technical standards. The binary logit model for the dichotomous dependent variable was appropriate, as was the use of Fairlie decomposition to analyze differences in satisfaction between groups within a nonlinear model framework. Instrument reliability and validity tests demonstrated very adequate results (a Cronbach's alpha total of 0.94 and a KMO value above 0.5). Furthermore, the use of clustered standard errors at the subdistrict level improved the accuracy of statistical inferences. However, several technical aspects require clarification, particularly regarding the justification for setting satisfaction coding thresholds (a score of 1–3 for dissatisfied and 4–5 for satisfied), as well as potential sampling bias due to the combination of online and offline surveys and the relatively long data collection period.

The conclusions presented are generally consistent with the empirical results. The key finding that policy factors—such as work incentives, social security, and cultural development—are more influential than demographic characteristics is supported by the results of logit regression and Fairlie decomposition. The use of quantitative data supplemented by qualitative interviews also strengthens the validity of the interpretation. In terms of research ethics and academic integrity, this study meets applicable standards, including the application of informed consent, respondent anonymity, and minimal risk to participants.

However, there are a number of key issues that need to be seriously addressed before this manuscript can be considered for publication.

First, regarding the conceptualization and measurement of policy satisfaction. Although the theoretical framework used draws on expectancy-performance theory and identity-based mechanisms, the operationalization of the dependent variable is based solely on a single global question regarding whether the policy “encourages local development.” This approach is less in line with the multidimensional nature of policy satisfaction, which encompasses cognitive evaluations, affective responses, and perceptions of distributive justice. The authors need to provide a strong theoretical justification for why a single-item measurement is considered adequate, or—if possible—reconstruct a composite satisfaction index based on available questionnaire items.

Second, there are potential issues of endogeneity and common method bias because the dependent and main independent variables are both measured through respondents' perceptions in a single survey instrument. The current manuscript does not acknowledge or discuss this limitation, nor does it implement methodological mitigation measures. Authors should explicitly address this potential bias and explain its implications for coefficient estimation and interpretation of results, or include relevant robustness checks where possible.

Third, interpretations of psychological mechanisms such as identity recognition, expectancy formation, and social comparison have not been supported by direct measurement or modeling of these constructs. Therefore, claims regarding these mechanisms should be clearly distinguished between empirical findings and theoretical interpretations. Authors are advised to formulate these mechanisms as plausible explanations, not as empirically confirmed mechanisms.

Fourth, the relatively limited sample size (N = 243) becomes even more problematic when the data are subdivided into subgroups based on age and region. Small subsamples, particularly in the elderly group, produce very large regression coefficients with wide standard errors , indicating potential instability of the estimates. Authors should address the issue of statistical power , provide robustness tests where possible, and provide explicit warnings about the limitations of interpreting the results of subgroup analyses.

Fifth, presenting logit regression results solely as logit coefficients limits readability and policy relevance, especially for readers across disciplines. The addition of marginal effects or predictive probabilities is highly recommended to clarify the substantive meaning and magnitude of policy impact.

Sixth, given the central role of Fairlie decomposition in drawing conclusions, authors should acknowledge the method's sensitivity to model specification, choice of comparator, and variable ordering. Including sensitivity analysis or alternative specifications would significantly enhance the credibility of the results.

In addition, there are several minor issues that also need attention, including the tendency to generalize findings outside the institutional and political context of China, the lack of transparency regarding the procedures for selecting respondents and analyzing qualitative data, inconsistencies in economic terminology, and descriptive redundancy in the background section of the study area.

Overall, this paper addresses an important topic and uses a relevant quantitative approach. However, key issues related to measurement validity, potential endogeneity, estimation stability, and interpretive caution require thorough addressing. With substantial revisions to these aspects, this paper has the potential to make a strong and valuable contribution to the academic literature.

Reviewer #2: This manuscript addresses an important policy issue and is generally well designed. The survey data and the use of logit regression and Fairlie decomposition are appropriate, and the main findings are supported by the results. The paper is clearly written and easy to follow.

The main issue concerns interpretation. Some conclusions, especially those related to 'collective identity' and 'social comparison,' are not directly measured in the statistical models and should be clearly presented as interpretive or qualitative explanations rather than as direct statistical results. The conclusions would also be stronger if key claims were more clearly linked to specific tables, such as the coefficients in Table 3 and the decomposition results in Table 7.

There are also minor statistical points that need clarification. The manuscript reports clustered standard errors at the Subdistrict level, but the analysis compares only two Subdistricts, which may affect the reliability of significance levels. In addition, the type of R² reported for the logit models should be clarified, and some subgroup results appear unstable due to small sample sizes.

No ethical or publication concerns are identified. With clearer interpretation, minor statistical clarification, and a more concise discussion, the paper would be improved and be ready for publication.

.

Reviewer #1: **Yes:**Misran MisranMisran MisranMisran MisranMisran Misran

Reviewer #2: No

---

## [Author Response · Author response to Decision Letter 1]

4 Mar 2026

Dear Editor and reviewers:

Thanks for your helpful recommendations. This is a revised manuscript with the title "Improving residents' satisfaction with administrative boundary changes: A comparative analysis based on the township-town merger policy "(PONE-D-25-65675). We have revised the manuscript thoroughly according to your comments and carefully proofread the manuscript to minimize typographical and grammatical errors.

Moreover, we attached the revised manuscript in the format of WORD for your approval. A document answering every question from you and the reviewers was also summarized and enclosed.

A revised manuscript with the correction sections red-marked was attached as the supplemental material for easy checking purposes.

Should you have any questions, please contact us without hesitation. Thank you again for your constructive suggestions!

---

## [Decision Letter · Decision Letter 1]

27 Mar 2026

Improving residents' satisfaction with administrative boundary changes: A comparative analysis based on the township-town merger policy

PONE-D-25-65675R1

Dear Dr. Zhao,

We’re pleased to inform you that your manuscript has been judged scientifically suitable for publication and will be formally accepted for publication once it meets all outstanding technical requirements.

Kind regards,

Tatchalerm Sudhipongpracha

Academic Editor

PLOS One

Additional Editor Comments (optional):

Reviewers' comments:

Reviewer's Responses to Questions

**Comments to the Author**

Reviewer #2: All comments have been addressed

2. Is the manuscript technically sound, and do the data support the conclusions?

Reviewer #2: Yes

3. Has the statistical analysis been performed appropriately and rigorously?

Reviewer #2: Yes

4. Have the authors made all data underlying the findings in their manuscript fully available?

Reviewer #2: Yes

5. Is the manuscript presented in an intelligible fashion and written in standard English?

Reviewer #2: Yes

Reviewer #2: - Authors still discuss identity mechanisms but now frame them as interpretive explanations rather than statistical outputs, which resolves the main conceptual problem.

- The empirical interpretation now clearly corresponds to the statistical results.

- Sample sizes are now transparent and readers can judge robustness.

- The issues that remains somewhat unresolved: 1) authors did not explicitly justify or discuss the limitation. However, this issue is not severe enough to require another revision round. 2) The table still reports R² values without specifying the type. However, this is very minor and does not affect interpretation.

.

Reviewer #2: No

---

## [Editor Report · Acceptance letter]

PONE-D-25-65675R1

PLOS One

Dear Dr. Zhao,

I'm pleased to inform you that your manuscript has been deemed suitable for publication in PLOS One. Congratulations! Your manuscript is now being handed over to our production team.

Kind regards,

on behalf of

Dr. Tatchalerm Sudhipongpracha

Academic Editor

PLOS One